# A quasi-experimental effectiveness evaluation of the 'Incredible Years Toddler' parenting programme on children's development aged 5: A study protocol

**Kate E. Mooney**[1,2]*, **Tracey Bywater**[1], **Sebastian Hinde**[3], **Gerry Richardson**[3], **John Wright**[2], **Josie Dickerson**[2], **Sarah L. Blower**[1]

**1** Department of Health Sciences, University of York, York, United Kingdom, **2** Bradford Institute for Health Research, Bradford Royal Infirmary, Bradford, United Kingdom, **3** Centre for Health Economics, University of York, York, United Kingdom

* kate.mooney@york.ac.uk

**Data Availability Statement:** There are no data to be shared with this manuscript as it is a study

## Abstract

Child behavioural and mental health problems have become a public health crisis. The consequences of poor mental health in childhood have large economic costs and consequences for the individual, their families, and for society. Early intervention through parenting programmes can reduce the onset of poor mental health in childhood, hence evaluating the effectiveness of parenting programmes is critical. The 'Incredible Years Toddler' parent programme is an education and training intervention designed to enhance the social and emotional wellbeing of children aged 1–3 years. Whilst previous studies show Incredible Years Toddler to provide promising effects on child outcomes in the short term, the research samples have lacked ethnic diversity and representation from socioeconomically deprived families. This quasi-experimental study is registered on ISRCTN (ISRCTN49991769). We will investigate the effectiveness of Incredible Years Toddler being delivered in three neighbourhoods in inner city Bradford, England. These neighbourhoods contain a socially and ethnically diverse population with 84% living in the poorest decile for England and Wales. Parents with a child aged 1–3 years old who are enrolled in Born in Bradford's Better Start interventional family cohort study are eligible for this study. Intervention participants will be matched to a demographically comparable control group using propensity score matching. This study will use retrospective and prospective data from participants who attended Incredible Years groups between September 2018 and April 2024. The required minimum sample is $n = 1336$ (ratio 1:3) to detect a small effect (odds = 1.5, $d = .20$) on the Early Years Foundation Stage profile total score at age 5; a measure of early child development that is routinely collected by teachers. We will also establish whether these effects are moderated by child age at entry to intervention, programme delivery mode, socioeconomic status, and ethnicity. We will also estimate the cost of the intervention and conduct a cost-consequence analysis.

protocol. In addition, the BiBBS data cannot be shared publicly as they are available through a system of managed open access. Before you contact us, please make sure you have read our Guidance for Collaborators. Our BiB Executive reviews proposals monthly and we will endeavour to respond to your request as soon as possible. You can find out about the different datasets in our Data Dictionary. If you are unsure if we have the data that you need please contact a member of the BiB team (borninbradford@bthft.nhs.uk). Once you have formulated your request please complete the 'Expression of Interest' form available here and send to borninbradford@bthft.nhs.uk. If your request is approved we will ask you to sign a Data Sharing Contract and a Data Sharing Agreement, and if your request involves biological samples we will ask you to complete a material transfer agreement.

**Funding:** This study has received funding from the National Lottery Community Fund (previously the Big Lottery Fund) as part of the A Better Start programme (Ref 10094849). SLB, TB, JD, SH, JW and GR are also supported by the National Institute for Health Research (NIHR) Yorkshire and Humber Applied Research Collaboration (ARC-YH; Ref: NIHR200166, see https://www.arc-yh.nihr.ac.uk,). The views in this publication are those of the authors and not necessarily those of the NIHR or the Department of Health and Social Care. The funders had no role in study design, data collection and analysis, decision to publish, or preparation of the manuscript.

**Competing interests:** SLB, TB, JD, SH, JW and GR were also supported by the NIHR Yorkshire and Humber Applied Research Collaboration (ARC-YH; Ref: NIHR200166,seehttps://www.arc-yh.nihr.ac.uk,). In addition, TB is a voluntary Trustee for Children's Early Intervention Trust (CEIT). Early Intervention Wales Training (EIWT) is owned by CEIT and offers training courses, previously including Incredible Years®(IY). Trustees do not benefit financially from trainings or CEIT/EIWT activities. This does not alter our adherence to PLOS ONE policies on sharing data and materials.

# Introduction

Child behavioural and mental health problems have become a public health crisis. One in eight children (aged 2–19 years) in England are estimated to experience a mental health disorder [1]. The prevalence of this increased with the COVID-19 pandemic, with the incidence rising from 11% in 2017, to 16% in July 2020 for children aged 5–16 years [2] In 2022, 18% of children aged 7 to 16 years were estimated to have a probable mental disorder [3]. Further, mental health problems are exacerbated for children living in socioeconomically disadvantaged circumstances [4, 5].

When children with mental health issues reach adulthood, they are more likely to be unemployed and to have contact with the criminal justice service [6]. Poor child mental health has large economic costs and consequences for both the individual and for society, for example, the average annual costs to mental health services for 12–15-year-olds with conduct disorders are £1,789, and £1,353 for those with emotional disorders [6].

The first three years of a child's life are a crucial time for their development, with disruptions in early neurodevelopment being a significant risk for development of later mental health problems [7, 8]. Early intervention through parenting programmes has the potential to prevent or reduce the personal and societal costs of poor child mental health [9], since some child behavioural and mental health issues may be attributable to insecure attachments and low parental sensitivity [10, 11]. Whilst group parenting programmes have been found to be effective for reducing child conduct problems and improving child socio-emotional wellbeing for parents of children aged three years or older [12, 13], evidence remains weak for the short and long-term effectiveness of parenting programmes for parents of children in the critical first three years of life, as well as in specific subgroups of parents [9, 12, 14].

## The incredible years toddler programme

One such series of programmes that may address this paucity of evidence are The Incredible Years (IY) parent programmes (www.incredibleyears.com), which are parent education and training group-based interventions informed by social learning theory and designed to enhance the social and emotional wellbeing of children aged 0–12 years. These manualised programmes are typically delivered by trained facilitators to groups of 10–12 parents for two hours a week for 12–20 weeks. This protocol outlines our plans to evaluate the Incredible Years Toddler (IY-T) programme for parents of 1-3-year-olds. In IY-T, parents learn effective strategies to help the child learn how to better manage their emotions and behaviour [15]. Whilst IY parent programmes for older children have evidence of positive short-term outcomes [14, 16], the more recently developed IY-T has less evidence in comparison.

There are only two previous studies which have investigated the effectiveness of IY-T [17, 18]. Perrin et al. [2014] conducted an RCT with 89 parents in the USA, and delivered IY Toddler through primary care (paediatric practices), rather than community settings. Significant improvements were found in parenting practices and child disruptive behaviours. Hutchings et al. [2017] conducted a community based RCT with 153 parents in Wales, and relied on geographical targeting to disadvantaged 'Flying Start' areas to deliver IY Toddler. Significant improvements were found in child development, home environment, and parental depression. In addition, a recent RCT of the 'E-SEE Steps' model (a proportionate universal approach delivering various intensities of the IY Baby and IY-T) found no significant differences in parent and child outcomes 18 months after randomisation [19]. However, the E-SEE trial was not designed or powered to establish the effectiveness of IY-T as a standalone programme.

It is of interest to know which sociodemographic groups of parents may benefit most from parenting programmes. Ethnic minorities represented 29% of the study population of the trial

based in the USA [18] and no information regarding ethnicity was reported for the study based in Wales [17]. With regards to socioeconomic status, parents in the study based in Wales were living in a disadvantaged area, however, the recruited participants were less disadvantaged than parents in a comparable trial (Welsh Sure Start trial), where community disadvantage rates were lower [17, 20]. Parents in the study based in the USA had household incomes ranging between $20,000 and $100,000 [18]. The IY-T intervention therefore has limited evidence of effect in the most vulnerable populations such as those from an ethnic minority or living in areas of socioeconomic deprivation. A meta-analysis of IY parent programmes indicated that the intervention effects are not moderated by socioeconomic or ethnic groups for child conduct problems, suggesting that the programme is similarly effective for families from different backgrounds [21]. However, this meta-analysis only compared 'White' groups to all other ethnic minorities. This means we cannot be certain of the meaning of this finding for different ethnic minority groups, since there may be much heterogeneity regarding the programme effectiveness across different ethnic minority groups. It would, therefore, be beneficial to test whether the finding that IY parent programmes are consistently effective across different ethnic groups generalises specifically to IY-T, and generalises to other long-term outcomes.

Due to the high cost of collecting long-term data, effectiveness trials are not usually funded to explore long-term outcomes over 12 months after the intervention took place. The aforementioned evaluations of IY-T examined parent and child outcomes up to 12 months after baseline [17, 20]—which is relatively long-term compared to other effectiveness trials, however, more long-term outcome studies are needed to explore ongoing effectiveness as well as economic and societal impacts. There is some evidence that children whose parents received the IY 'parent training and child treatment' programme maintain positive behavioural improvements for oppositional defiant disorder and conduct disorder several years post-follow-up [22], however, studies are needed to confirm the long-term results of all IY interventions for various ages, diagnoses, and demographic populations [14].

## Methods for evaluating child and family preventative services in practice

Whilst RCTs are the most rigorous method of establishing the effectiveness of an intervention [23], they are susceptible to several issues [24, 25]. RCTs can suffer from a lack of generalisability, as participants that volunteer to participate might not be representative of the population being studied. Parents who are at the point of 'readiness to change' are unlikely to agree to take part in a study in which they might be randomised not to get help, resulting in recruitment bias [23, 25]. A further problem is that it is usually not possible to control the opportunities through which a parent can access an intervention; since they can be made available through third sector organisations, the internet or through books. A parent who has heard about a programme during their recruitment to a study may decide to enrol in the programme through another route (even if they are in the control group of the RCT) [25]. Finally, RCTs are susceptible to a high number of participants being lost to follow up [23], especially when a study aims to explore long term effects of the intervention.

A quasi-experimental study cannot provide the level of certainty in establishing causal effectiveness that an RCT does, as it lacks random assignment to treatment group [24, 26]. However, it has advantages in terms of efficiency and pragmatism, and can overcome the RCT design limitations outlined above, such as providing ecological validity by evaluating an intervention when delivered in its natural (non-trial) context [24]. This quasi-experimental study evaluates IY-T as delivered in the Better Start Bradford programme by Barnardo's. Parents at the point of 'readiness to change' are free to enrol on this intervention immediately (or not),

thus reducing recruitment bias. The participants are enrolled in the Born in Bradford's Better Start (BiBBS) interventional family cohort study, which runs in parallel to Better Start Bradford [27]. By utilising the BiBBS cohort, this study has linkage to a routinely collected and long-term outcome, and this mitigates the risk of losing participants to follow up. Finally, the BiBBS cohort contains an ethnically diverse population (61% Pakistani heritage; 12% White British; 8% other South Asian and 6% Central and Eastern European ethnicity), with the majority (84%) living in the lowest decile of the Index of Multiple Deprivation [28]. This study therefore provides a unique opportunity to advance our understanding of the effectiveness of IY-T in an ethnically diverse and socioeconomically deprived population.

## Rationale and objectives for this study

The long-term effectiveness of IY-T has not yet been investigated using a predominantly disadvantaged and ethnically diverse cohort sample with routinely linked data. This study protocol outlines our plans to be the first to provide insights into the longitudinal effectiveness of IY-T on children's social-emotional development and early years school skills at 5 years, approximately 2–4 years after their parents' attendance at IY-T groups. The study objectives are to:

1. Establish if IY-T is effective in improving child social and emotional development at age 5 (primary outcome) when compared to services as usual

2. Establish if IY-T is effective in improving children's early years school skills at age 5 (secondary outcome) when compared to services as usual

3. Establish whether the impact of IY-T on the above outcomes is moderated by (a) child age at time of intervention, (b) programme delivery mode (online versus face to face), and (c) socioeconomic status

4. Estimate the cost of the intervention (including delivery and training to deliver IY-T) and combine them with the effectiveness evidence through a cost-consequence analysis.

## Methods

### Registration

We have used the STROBE (2021) guidelines for reporting methods in this protocol and will use the guidelines for the publication of our results. This study has been registered retrospectively on ISRCTN, a clinical database for RCTs and effectiveness studies (ISRCTN49991769).

### Design

This effectiveness evaluation will use a quasi-experimental design. A quasi-experimental design lacks random assignment and uses self-selection (by which participants choose treatment or a lack of treatment for themselves) [26]. The analyses will use propensity score matching, where multiple control cases will be matched to intervention cases (see statistical methods for further details).

### Setting

This study is set in Bradford; a city in Northern England with high levels of socioeconomic deprivation and ethnic diversity [29]. IY-T is being delivered by Barnardo's as part of the Better Start Bradford initiative (https://www.barnardos.org.uk/what-we-do/services/incredible-years-happy; https://www.betterstartbradford.org.uk/). Better Start Bradford is an initiative

funded by the National Lottery Community Fund (previously the Big Lottery fund), which provide services for expectant families and families with children aged 0–3 in Bowling and Barkerend, Bradford Moor and Little Horton (three areas within Bradford).

Born in Bradford's Better Start (BiBBS) is an interventional birth cohort study, which runs in parallel to Better Start Bradford, and is designed to evaluate the effectiveness of Better Start Bradford's early life interventions [27]. BiBBS recruits pregnant women and their babies who live in the Better Start Bradford areas, regardless of intervention participation. Women complete an in-depth baseline questionnaire during pregnancy, and consent to routine linkage to their and their child's health and education records and Better Start Bradford intervention participation. BiBBS recruits 54% of the eligible population and is generally representative of the local pregnant population, therefore the intervention and control participants in this study are also representative of this area [28].

## Ethical and safety considerations

The informed consent procedure in BiBBS includes women providing written informed consent to the evaluation of Better Start Bradford interventions using quasi-experimental methods for themselves and their child, including the use of cohort participants to create intervention and control groups, and use of routine data to evaluate interventions.

The protocol for BiBBS recruitment and collection of routine outcome data was approved by Bradford Leeds NHS Research Ethics Committee (15/YH/0455). Research governance approval was gained from Bradford Teaching Hospitals NHS Foundation Trust (BTHFT). The sponsor is the BTHFT Research Management and Support Office who are authorised to conduct independent auditing of sponsored studies. The existing ethics includes approval for the evaluation of Better Start Bradford interventions using quasi-experimental methods, including the use of cohort participants to create control groups.

IY-T is delivered as a part of usual practice. Ethical considerations with regards to the delivery of the programme can be found in the IY-T service design plan, and issues arising are managed by the service delivery team and Better Start Bradford as service commissioners, in accordance with local guidance, e.g., such as safeguarding procedures, access to appropriate mental health support. Key members of the research team are trained in good clinical practice. There are unlikely to be harms to individuals from taking part in the evaluation as it is non-invasive, and the practitioners delivering the intervention are trained in safeguarding.

## Participants

**Eligibility criteria.** To be eligible for this evaluation, participants must be enrolled in the BiBBS cohort. Whilst not all IY-T attendees will be BiBBS participants, all observations in the study sample, i.e., analysed intervention and matched comparison group, will be BiBBS participants.

Eligibility criteria for the intervention group are:

- Consented to participate in BiBBS

- Parent consented to a referral and enrolled in IY-T

  Eligibility for the matched comparison group are:

- Consented to participate in BiBBS

- Parent of a child aged between 12 and 36 months at any time in the duration of service delivery (to ensure that matched cases are comparable)

- Parent has not been referred or enrolled in IY-T at any timepoint

- Parent has been matched to an individual in the intervention group (see statistical methods section)

**Intervention.** IY-T is the intervention. Parents living in the BSB area with a child aged 1–3 are eligible to enrol in IY-T. IY-T receive most of their enrolled participants through their own engagement with the community directly, and also through engaging with schools and nurseries. IY-T aim to recruit approximately 12 parents to each group that they run.

Within IY-T, parents learn how to help their toddlers feel loved and secure, encourage social and emotional development, and establish strategies for developing routines, handling separation, and managing misbehaviour [15]. Parents/carers receive 3 promotional contacts prior to the beginning of the group via assertive outreach, consisting of telephone contact and at least one home visit. The initial telephone contact introduces the parents to the project and Group Facilitators, and aims to build the participants confidence in attending. The home visits are intended to create a sense of rapport between the family and Group Facilitators and alleviate any barriers that families might have in accessing the group such as crèche, language difficulties, or concerns about what the group might involve. The intervention will be delivered in a combination of face-to-face and virtual formats (dependent on lockdown rules in place during the COVID-19 pandemic at the time of the study and participant needs) by trained Group Facilitators who receive regular supervision from an accredited IY mentor.

The 13 IY Toddler Basic sessions cover eight topics:

- Part one –Child-Directed Play Promotes Positive Relationships

- Part two–Promoting Toddler's Language with Child-Directed Coaching

- Part three–Social and Emotion Coaching

- Part four–The Art of Praise and Encouragement

- Part five–Spontaneous Incentives for Toddlers

- Part six–Handling Separations and Reunions

- Part seven–Positive Discipline-Effective Limit Setting

- Part eight–Positive Discipline-Handling Misbehaviour

Further detail on the content and delivery of the IY-T intervention, and materials, can be found here: https://incredibleyears.com/programs/. The IY-T theory of change and logic model developed by Barnardo's for Bradford delivery states that IY-T will result in improvements in child social and emotional development over the medium and long term. The logic model is provided in a S2 File.

## Outcome (early years foundation stage profiles)

The primary and secondary outcomes are obtained through the child's Early Years Foundation Stage Profile Profile (EYFSP) which will be linked to individual BiBBS children through their routine education records. The EYFSP is completed by teachers for children when they reach age five, at the end of their last term in reception. It contains 17 Early Learning Goals (ELGs) relating to different areas of learning. Previously, children were assigned a score of either "Emerging" (not reaching expected levels), "Expected" (reaching expected levels), or "Exceeding" for each ELG [30, 31]. For the new EYFSP (post September 2021), children are assigned a

**Table 1. Overview of the early learning goals (ELGs) in the old and new versions of the Early Years Foundation Stage Profile (EYFSP).**

| Area of Learning | "Old" EYFS ELGs (See Section 6 of 2021 EYFSP handbook for further detail) | "New" EYFS ELGs (See Section 3.4 of 2021 EYFSP early adopter handbook for further detail) |
|---|---|---|
| **Communication and language development (primary outcome)** | **1. Listening and attention**<br>**2. Understanding**<br>**3. Speaking** | **1. Listening, attention and understanding**<br>**2. Speaking** |
| Physical development | 4. Moving and handling<br>5. Health and self-care | 3. Gross motor skills<br>4. Fine motor skills |
| **Personal, social and emotional development (primary outcome)** | **6. Self-confidence and self-awareness**<br>**7. Managing feeling and behaviour**<br>**8. Making relationships** | **5. Self-regulation**<br>**6. Managing self**<br>**7. Building relationships** |
| Literacy | 9. Reading<br>10. Writing | 8. Comprehension<br>9. Word reading<br>10. Writing |
| Mathematics | 11. Numbers<br>12. Shape, space and measures | 11. Number<br>12. Numerical patterns |
| Understanding the world | 13. People and communities<br>14. The world<br>15. Technology | 13. Past and present<br>14. People, culture and communities<br>15. The natural world |
| Expressive arts and design | 16. Exploring and using media and materials<br>17. Being imaginative | 16. Creating with materials<br>17. Being imaginative and expressive |

Note: 1. For a detailed description of what is expected of a child within an ELG, see GOV.UK [2021] [32].

2. The bolded ELG's are the goals used for the primary outcome.

score of either "Emerging" or "Expected" for each ELG. See Table 1 for an overview of both the new and old EYFS ELGs.

The EYFSP is an important predictor of later academic attainment and developmental difficulties [33, 34]. Whilst previous studies and The Department for Education have used the 'good level of development' (GLD) measure, this is a binary outcome and reduces all goals into one composite measure. An alternative to the GLD is to instead assign numerical scores to each category in the EYFSP (e.g., zero for emerging, one for expected, and two for exceeding), and sum these scores into a 'total score' composite (ranging between 0–34).

For each child, every goal met (or exceeded) will be assigned a one, and each goal scored as 'emerging' will be assigned a zero. The outcomes will be summed in the following ways:

1. For the primary outcome, we will examine only the most relevant ELG's to child social and emotional development. These are the 'communication and language' and 'personal, social, and emotional' ELGs, with total 5 goals (see Table 2). The binary responses for each of these 5 goals will be summed to generate a summary score between 0 and 5.

2. For the secondary outcome (child early years development), the binary responses for each of the 17 ELGs will be used to generate a summary score, where a total score between 0 and 17 will indicate how many of all the ELGs were assigned a score of 'expected'.

Participants who entered reception prior to or during September 2020 will be assessed according to the old version (though this is anticipated to be a small number of participants), and participants in IY-T who enter reception after or during September 2021 will be assessed according to the new version. The measures will be made comparable by collapsing all participants who met or exceeded goals to be assigned a one, and each goal scored as emerging will be assigned a zero.

**Table 2. Overview of the covariates (matching variables, effect moderators, and covariates) to be included in model.**

| Covariates (measurement tool if relevant) | Measurement | References providing justification |
|---|---|---|
| Matching variables for propensity score | | |
| Prenatal attachment (Prenatal Attachment Inventory) | Continuous | [10, 11, 41] |
| Prenatal Depression (PHQ-9) | Continuous | [10, 11, 41, 42] |
| Prenatal Anxiety (GAD-7) | Continuous | As above |
| Ethnicity | Categorical | [10, 11, 30, 41, 43] |
| Socioeconomic position (financial security) | Continuous | [30, 42, 44–46] |
| Speaks English | Binary | [30, 46] |
| Immigration status (proportion of life spent in UK) | Continuous | [30, 47, 48] |
| Length of time in COVID lockdown | Continuous | [49] |
| Moderators | | |
| Child age | Continuous | N/A |
| Programme delivery mode (face to versus or virtual) | Binary | N/A |
| Socioeconomic position (financial security) | Continuous | N/A |
| Ethnicity | Categorical | N/A |

## Sample size calculations

This study has a fixed sample size dependent on the projects capacity and recruitment rates, and this section explores the required sample size to obtain 80% power. The EYFSP will be modelled as an ordinal outcome, using a proportional odds model. We have calculated the numbers at which we will have 80% power to detect an odds ratio of 1.5, which is thought to be equivalent to a Cohen's *d* of .20 (small effect size) [35]. We have used an allocation ratio of 1:3 (intervention:control). The Stata command *artcat* was used for all sample size calculations.

1. For the primary outcome using five ELGs, we will be powered to detect a difference with 1336 participants total (334 intervention participants and 1002 control participants).

2. For the secondary outcome using all 17 ELGs, we will be powered to detect a difference with 950 participants total (237 intervention participants and 713 control participants).

## Data timeline

This study will use all available data since IY-T began (1st September 2018) up until the end of their service delivery period (expected to be 30th April 2024). Fig 1 provides the participant pathway through BiBBS cohort recruitment, participation in IY-T (or non-participation), and school entry with the outcome collected during reception year. Parents may enrol in the intervention when their child is aged anywhere between 1 and 3 years old, which means outcome data may be collected anywhere between two and four years after intervention participation. The routine education outcome in the EYFSP, which is reported by teachers at the end of children's reception year in primary school (when child is aged 5 years old, equivalent to 60 months). The final analyses will be after August 2028, when the youngest children within the delivery period reach age 5 years.

## Statistical methods

A detailed statistical analysis plan will be preregistered on the Open Science Framework (OSF), this section is provided as a summary. Our analyses will use the propensity score matching method, which estimates the probability of a subject to receive a treatment conditional on the set of covariates. Matching patients with a similar estimated propensity score will

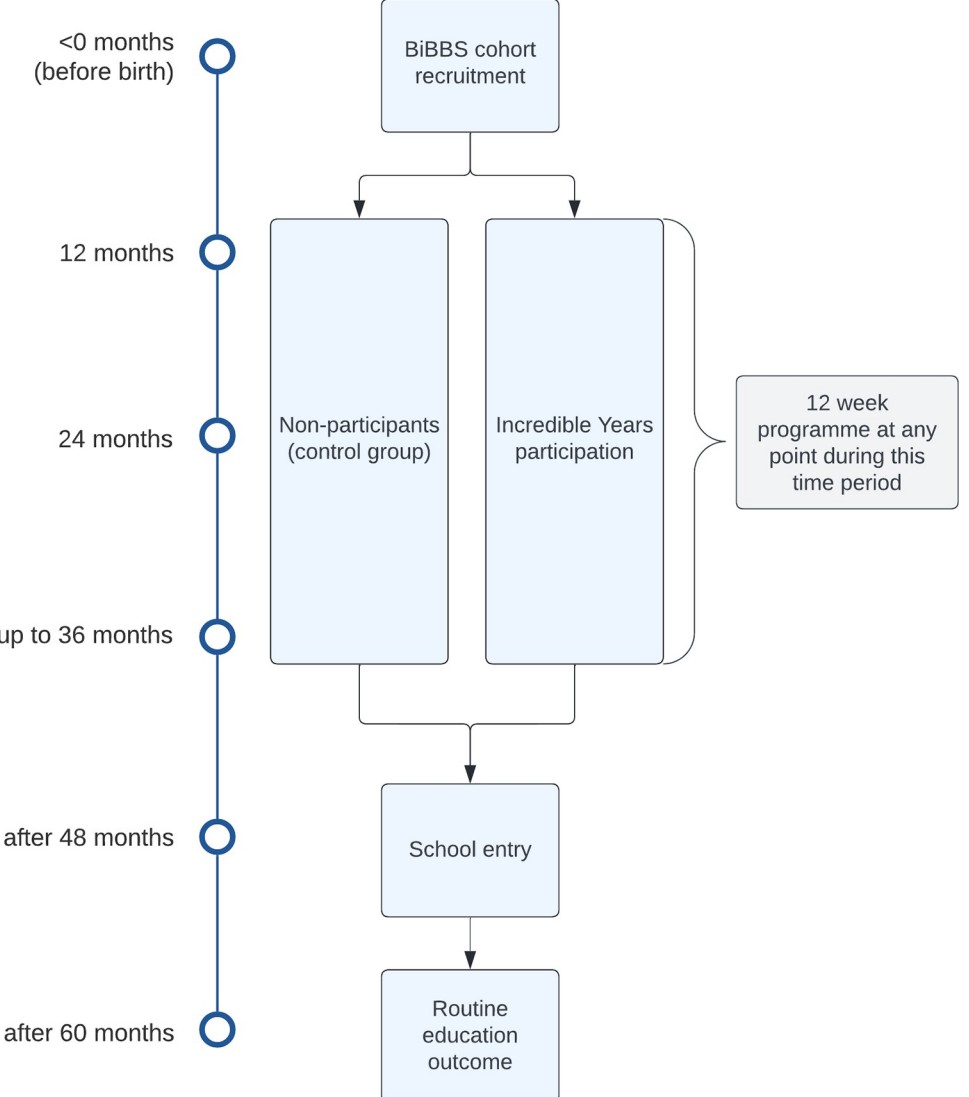

**Fig 1. Overview of timeline of participation and data collection.** Note: Caregivers are eligible for IY-T when their child is aged 12–36 months old. Participation in the IY-T programme lasts approximately 12 weeks.

create approximate balance for baseline characteristics [36]. The propensity score method will use one-to-*many* matching, by matching individual treatment cases to multiple control cases [37]. After matched groups have been created, ordinal logistic regression analyses will be run using group assignment as the independent variable, and the primary and secondary outcomes as the dependent variables.

## Variables

The Directed Acyclic Graph (DAG) in Fig 2 identifies the confounding variables that will be used as matching variables in the propensity score [38, 39]. We will match on the 7 characteristics on the left-hand side of the graph (prenatal attachment, prenatal depression, prenatal anxiety, ethnicity, socioeconomic status, language status, and immigration status) to estimate the effect of treatment (IY-T) on the primary outcome (child social and emotional development).

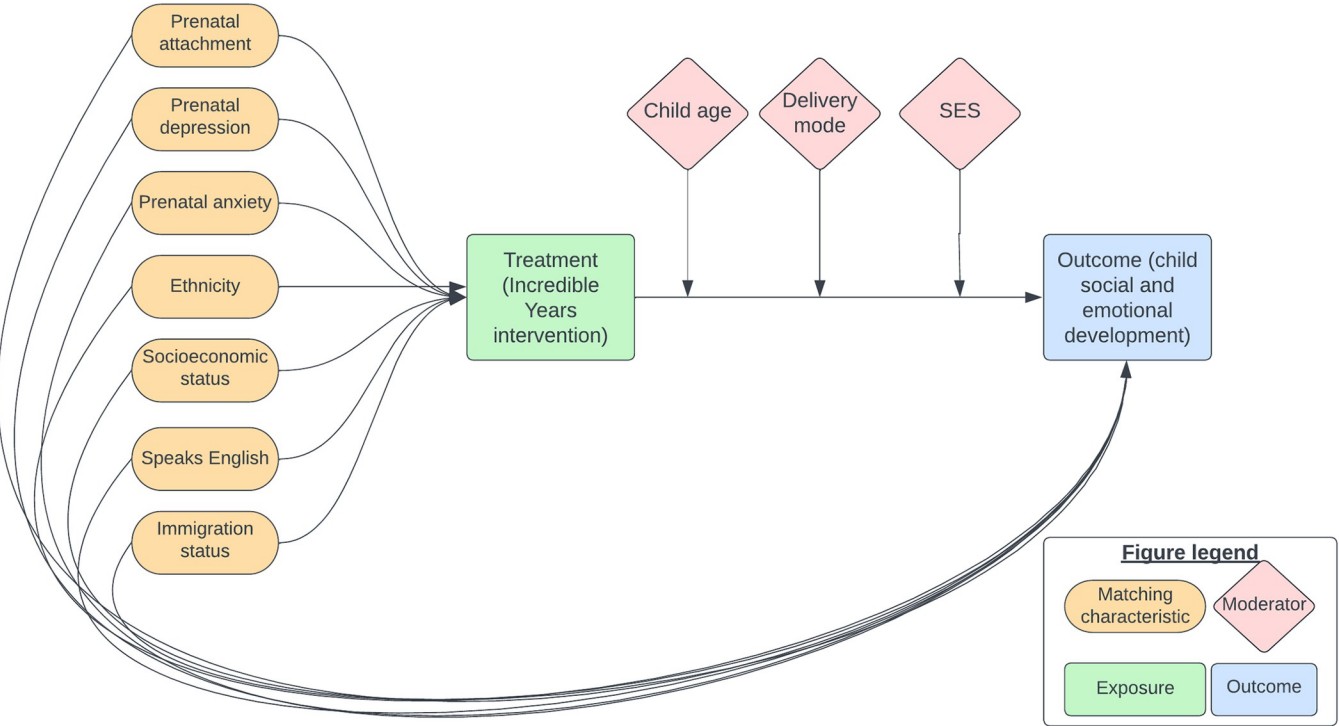

**Fig 2. Directed Acyclic Graph of associations between confounders, covariates, moderators, treatment exposure, and outcome.** Notes: 1. SES = socioeconomic status. 2. The secondary outcome (children's early years school skills) is not depicted here.

The justification for the selection of these variables is given in Table 2 below. Variables which affect program participation but not the outcomes should not be used in propensity score matching, as this increases the error in the estimated values [40]. Matching variables were therefore selected if they were thought to have an association with both uptake of treatment, and with the outcomes (child early years development), and there was evidence to suggest this was the case. See Table 2 for a summary of the evidence for using a matching variable, and the tested moderators of the effect. We have proposed that we will test the effects of four moderators: child age, programme delivery mode, family socioeconomic status, and ethnicity, however, this will only be possible if a sufficient sample size is reached.

All variables are obtained from the BiBBS baseline questionnaire, which takes place before the child is born (see https://borninbradford.github.io/datadict/bibbs/ for details).

## Included participants

We will conduct both Intention to Treat (ITT) analyses to ascertain the effects of enrolling in the intervention [50, 51], and per protocol analyses to ascertain the effects of receiving the treatment [52]. The ITT analyses will therefore include all who enrolled in the intervention after an initial home visit (even if they did not go on to participate in it), and the per protocol analyses will only include 'completers', defined as those who take part in at least 8 out of the 13 sessions. This will give us more accurate estimates of the average treatment effect on the treated participants.

Parents are eligible for the evaluation regardless of whether they are the mother, father, or another type of co-parent. Whilst we anticipate that the majority of parents in this study will be mothers, we will describe the number of fathers and other co-parents if possible.

## Missing data methods

If missing data are higher than 5% in any of the matching variables or moderators, then multiple imputation will be used to handle missing data [53]. Missing data rules for within measures will be followed where available (e.g., within the PHQ-8). Counts of children missing the outcome will be presented with reasons where possible, and tabulations of those who are missing the outcome will be created by the matching variables to explore whether there are any differences in characteristics between those with outcome data and those without.

## Robustness testing

Histograms of the distribution of the EYFSP variable will be created and the analysis be run with the EYFSP as both a categorical and continuous variable. The model fit and parameters will be compared between the models. The Stata user written command 'overfit' will be used to measure the amount of overfitting generated by each model.

## Software

Stata-17 will be used for data handling and analyses. The packages *teffects* and *psmatch2* will be used for matching, and *ologit* will be used for regression analysis.

## Economic analysis

The objectives for the economic analysis are to:

1. Estimate the cost of the intervention (including delivery and training to deliver IY-T),

2. seek to combine these costs with the effectiveness data to inform a cost-consequence analysis, and

3. to explore extrapolating the effectiveness data generated by this analysis to longer-term outcomes through an existing simulation model.

The costing analysis will seek to apply an existing framework for conducting micro-costing [54] in addition to previous costings conducted in a previous randomised trial of IY (the E-SEE study) [19]. Where possible, we will generate a cost per participant in each group, estimate the cost of the intervention, and who that cost falls upon. These costs will be reported alongside the effectiveness outcomes described in the 'Outcome' sections of this protocol to inform a cost-consequence analysis, including potential sub-group analysis stratified by socio-economic status. Finally, by applying ongoing research mapping the EYFSP to the Strengths and Difficulties Questionnaire (SDQ) [Mooney et al., in preparation] to a life course dynamic microsimulation model [55] we will explore the extrapolation of findings of this study over the lifetime of the children who both receive IY-T and the matched control group.

## Pilot data

An interim profile of BiBBS for all pregnant women recruited between January 2016 and November 2019, with an expected due date between 1st April 2016 and 8th March 2020, indicated that the BiBBS cohort sample is representative of the Better Start Bradford pregnant population. Furthermore, a high proportion (85%) of BiBBS families had engaged in one or more of the Better Start Bradford interventions, including in IY-T [28]. This data demonstrates the feasibility of conducting the IY-toddler evaluation in a disadvantaged population.

## Process evaluation

A process evaluation will explore aspects of implementation fidelity including dose (number of sessions attended) and adherence (self-reported Group facilitator checklists). Parent and practitioner perspectives on the impact of IY-T, and barriers and facilitators to implementation will be explored. Fathers and other co-parents will be involved in the process evaluation where possible. The plans for this study will be specified in a separate process evaluation protocol.

## Data management

As per the BiBBS protocol, record matches for routine data linkage, including health, education and IY-T data will be validated based on NHS number plus multiple non-unique identifiers (e.g., surname, date of birth) where possible. Identifiable data is only accessible by a small number of the research team (not the authors of this study), and is only done for approved processes such as withdrawals; address updates etc. Data for evaluations is processed and fully anonymised before being shared with researchers. The central database, hosted by BTHFT, will store data obtained from all sources. Data from each source will be linked at the BiBBS person level and will be structured and maintained by BiBBS data managers as a long-term strategic store to service cohort data capture and analysis. The entire database schema and data will be backed up nightly. All data will be stored within the cohort-specific Relational Database Management System (RDBMS) and managed using Microsoft SQL Server 2018 Management Studio.

# Discussion

This quasi-experimental evaluation will establish the effectiveness (or not) of the IY-T programme on children's social and emotional wellbeing and early years school skills two to four years post parent attendance in IY-T groups. It will provide a unique insight into the effects of the IY-T programme in an ethnically and socioeconomically diverse population. This study will harness the BiBBS cohort to enable links to routine data, allowing an exploration of the effects of IY-T on children's EYFSP scores. The intervention is offered in a real-world context, in a setting where the intervention would exist whether we evaluated it or not. We will utilise an existing data set to conduct a quasi-experimental evaluation; hence it allows for a low-cost, efficient, and timely evaluation compared to a full RCT. The findings regarding intervention effectiveness of IY-T will be generalisable to other urban areas that are ethnically diverse with high levels of socioeconomic deprivation. They may also be generalisable beyond IY-T to other parenting programmes more widely, though this will depend on the contents and implementation of the individual programmes.

Beyond this study, it will be possible to follow the very long-term outcomes of the children whose parents have and have not received IY-T in the BiBBS cohort. Whilst this planned study aims to first examine the impact of IY-T on children's social and emotional wellbeing and school skills at age 5, it will also eventually be possible to examine any impacts of IY-T into adolescence (e.g. through routine education data) and adulthood (e.g. through access to medical records to infer incidence of mental health problems). Again, this highlights the utility of a quasi-experimental study nested within an interventional cohort.

## Study limitations

There is a paucity of routinely collected data within the early years of a child's life, which limited our choice of an outcome measure. Whilst the EYFSP is a well-timed measure and

represents a construct of interest to this study, other measures relating to outcomes that IY-T aims to effect would have been beneficial (e.g. of parent-child bonding) [56]. Further, as collection of EYFSP data was paused during the COVID pandemic, there will inevitably be missing outcome data for a small number of children. However, it is worth noting that EYFSP and other routinely collected health and education outcome data is regularly used as the basis for most policy-making decisions and this study reflects real-world conditions.

A further limitation of this evaluation is that it will not consider the involvement of fathers and other co-parents in the IY-T groups. Whilst fathers and co-parents of children are eligible for both BiBBS and IY-T, the number of enrolled fathers and co-parents in both the cohort and IY-T is unlikely to be sufficient to conduct any meaningful analyses on these groups. Whilst it can be more challenging to recruit fathers to parenting research studies [57, 58], we acknowledge that much further research and efforts is needed on the involvement of fathers and co-parents in parenting interventions, and will aim to explore this in our process evaluation of IY-T.

## Conclusion

This study will contribute to the growing evidence base regarding the effectiveness of the IY-T programme [17–19], and more broadly, the literature regarding effectiveness of parenting education programmes in underserved communities [9, 12, 14, 21]. This quasi-experimental study will provide useful evidence for policy makers and commissioners regarding IY-T and parenting education programmes, and it may ultimately improve outcomes for children living in a disadvantaged area.

## Supporting information

**S1 File. Strengthening the reporting of observational studies in epidemiology statement checklist.**
(DOCX)

**S2 File. Logic model of incredible years (Toddler).**
(DOCX)

## Acknowledgments

The integration of research and practice in Bradford has only been possible because of the enthusiasm and commitment of staff and volunteers across children's services in Bradford. We are grateful to all Born in Bradford staff, the Better Start Bradford partnership and staff, all Better Start Bradford project teams, health professionals, local authority and voluntary and community sector organisations who have supported the integration of research into practice. We are grateful to all the families taking part in BiBBS and all members of the Community Research Advisory Group.

## Author Contributions

**Conceptualization:** Kate E. Mooney, Tracey Bywater, John Wright, Josie Dickerson, Sarah L. Blower.

**Methodology:** Kate E. Mooney, Tracey Bywater, Sebastian Hinde, Gerry Richardson, Josie Dickerson, Sarah L. Blower.

**Project administration:** Kate E. Mooney.

**Writing – original draft:** Kate E. Mooney.

**Writing – review & editing:** Kate E. Mooney, Tracey Bywater, Sebastian Hinde, Gerry Richardson, John Wright, Josie Dickerson, Sarah L. Blower.

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
