## [Decision Letter · Decision Letter 0]

10 Aug 2023

PONE-D-23-16914A quasi-experimental effectiveness evaluation of the 'Incredible Years Toddler’ parenting programme on children’s development aged 5: A study protocolPLOS ONE

Dear Dr. Mooney,

Thank you for submitting your manuscript to PLOS ONE. After careful consideration, we feel that it has merit but does not fully meet PLOS ONE’s publication criteria as it currently stands. Therefore, we invite you to submit a revised version of the manuscript that addresses the points raised during the review process. As Academic Editor I would like to apologise for the delay in review.  It has proven difficult to obtain available reviewers as we moved into the summer months.  The manuscript has received one review and I have also reviewed the manuscript.  I'd like to thank you for such a high standard of submission.  The reviewer has highlighted some minor changes to the manuscript and having reviewed I have nothing to add in addition to those suggested.   

We look forward to receiving your revised manuscript.

Kind regards,

Charlotte Lennox

Academic Editor

PLOS ONE

Journal Requirements:

3. Please expand the acronym “NIHR” (as indicated in your financial disclosure) so that it states the name of your funders in full.

"This study has received funding from the National Lottery Community Fund (previously the Big Lottery Fund) as part of the A Better Start programme (Ref 10094849). The funder was not involved in the design of the study and collection, analysis, and interpretation of data nor in writing the manuscript.

SLB, TB, JD, SH, JW and GR are also supported by the NIHR Yorkshire and Humber Applied Research Collaboration (ARC-YH; Ref: NIHR200166, see https://www.arc-yh.nihr.ac.uk,). The views in this publication are those of the authors and not necessarily those of the NIHR or the Department of Health and Social Care." 

"SLB, TB, JD, SH, JW and GR were also supported by the NIHR Yorkshire and Humber Applied Research Collaboration (ARC-YH;Ref: NIHR200166,seehttps://www.arc-yh.nihr.ac.uk,). In addition, TB is a voluntary Trustee for Children’s Early Intervention Trust (CEIT). Early Intervention Wales Training (EIWT) is owned by CEIT and offers training courses, including Incredible Years®(IY). Trustees do not benefit financially from trainings or CEIT/EIWT activities."

7. We note you have included a table to which you do not refer in the text of your manuscript. Please ensure that you refer to Table 1 in your text; if accepted, production will need this reference to link the reader to the Table

8. Please upload a new copy of Figures 1 and 2 as the detail is not clear. Please follow the link for more information: " ext-link-type="uri" xlink:type="simple">https://blogs.plos.org/plos/2019/06/looking-good-tips-for-creating-your-plos-figures-graphics/"
https://blogs.plos.org/plos/2019/06/looking-good-tips-for-creating-your-plos-figures-graphics

Reviewers' comments:

Reviewer's Responses to Questions

**Comments to the Author**

1. Does the manuscript provide a valid rationale for the proposed study, with clearly identified and justified research questions?

Reviewer #1: Yes

2. Is the protocol technically sound and planned in a manner that will lead to a meaningful outcome and allow testing the stated hypotheses?

Reviewer #1: Yes

3. Is the methodology feasible and described in sufficient detail to allow the work to be replicable?

Reviewer #1: Yes

4. Have the authors described where all data underlying the findings will be made available when the study is complete?

Reviewer #1: Yes

5. Is the manuscript presented in an intelligible fashion and written in standard English?

Reviewer #1: Yes

6. Review Comments to the Author

You may also provide optional suggestions and comments to authors that they might find helpful in planning their study.

Reviewer #1: There are too few longer term follow-up studies of early intervention programmes. This an interesting, pragmatic, study that makes use of publicly held data on children enabling longer term follow-up of the effectiveness of the IY-T programme to children recruited at ages 1 – 3. It describes the plan to report on school readiness at 5 years of age using the Early Years Foundation Stage Profile It is excellent that the authors have preregistered their plans and submitted this paper.

Comments

The paper needs a careful re-read of the grammar, for example it is not good English to use the same word twice in a sentence. p 10 participants must be participating and in the following sentence participants occurs three times.

All numbers under 10 should be written as words unless age

Presume it is the “FIRST” term in reception P12

Stick to IY-T once it is introduced rather than Incredible Years p14

The Welsh IY-T study was working with a vulnerable population just that it was not as vulnerable as in a trial where participants were recruited based on individual characteristics rather than just targeting a high challenge area.

Not sure I understand the point being made on P5 “It would therefore be beneficial……”

I am not a statistician so cannot comment on the proposed analysis plan but think it would be helpful to add into the text the 7 (seven) characteristics p 15/16

Unfortunately your sample is small to undertake comparisons of the live versus remote delivery

Will the matching of characteristics to create the “control” group also include child age at time of intervention as that affects length of follow-up?

P18 is it 12 or 13 sessions?

I think it is a limitation that this only relates to mothers and wonder whether fathers were included in the groups and what data is available on them

I think it would be helpful to say a little more about the theory of change model

Really not sure of the potential of the results to generalise to other programmes. which can be varied in content, process and access.

TB declararation

The Children’s Early Intervention Trust is not a training centre for IY programmes and has not been since 2019.

7. PLOS authors have the option to publish the peer review history of their article (what does this mean?). If published, this will include your full peer review and any attached files.

Reviewer #1: No

---

## [Author Response · Author response to Decision Letter 0]

23 Aug 2023

We thank the reviewer for their useful comments and review of our manuscript. We are pleased that the reviewer sees the study as interesting, and that they acknowledge the benefits of publishing the protocol for this study. 

The paper needs a careful re-read of the grammar, for example it is not good English to use the same word twice in a sentence. p 10 participants must be participating and in the following sentence participants occurs three times. 

Thank you for highlighting that the manuscript needed a careful re-read. We have done this and made changes throughout. Specifically, we have made changes on p.10 to instead refer to ‘enrolled’ in the BiBBS cohort and ‘observations’ in the sample, and feel this is now clearer. 3-22

All numbers under 10 should be written as words unless age 

We have now made this change throughout the manuscript. Specifically, we have made this change on page 11 where we refer to the eight topics in the programme, and on page 13 where we refer to the number of goals in the EYFSP. We have not made this change when we refer to figures or tables to adhere to PLOS ONE’s style requirements. 

Presume it is the “FIRST” term in reception P12 

It is the last term at the end of their reception year, we have now updated this. 

Stick to IY-T once it is introduced rather than Incredible Years p14 

Thank you for noticing this. We have now updated this to refer to IY-T throughout. 

The Welsh IY-T study was working with a vulnerable population just that it was not as vulnerable as in a trial where participants were recruited based on individual characteristics rather than just targeting a high challenge area.. 

Thank you for highlighting this. We have amended the text where we describe this study, and feel it is now clearer that parents were recruited based on living in a disadvantaged area. 

Not sure I understand the point being made on P5 “It would therefore be beneficial……” 

Thank you for noting that this sentence was not clear. We have now made a clarification in the text to make it clear that we are referring to the finding that IY parent programs are effective across different ethnic groups. 

I am not a statistician so cannot comment on the proposed analysis plan but think it would be helpful to add into the text the 7 (seven) characteristics p 15/16 

We have now noted the seven matching characteristics in the text, and can see that this is a helpful clarification for readers of the study. 

Unfortunately your sample is small to undertake comparisons of the live versus remote delivery 

We have not given an expected sample size in the manuscript as this is not known at present. The sample size depends on how many parents enrol in IY-T in the coming years, which we are not able to predict. However, we agree with the reviewer that moderation analyses will only be possible if a sufficient sample size is reached, and have now clarified this in the text. 

Will the matching of characteristics to create the “control” group also include child age at time of intervention as that affects length of follow-up? 

We agree with the reviewer that exploring effects by child age is important, as this will be associated with the length of follow-up time for outcome. However, child age is not included as a matching characteristic in this study, as we do not believe that child age will effect program participation in this sample. For propensity score matching, matching characteristics should effect both programme uptake and the outcome(s). Instead, we have decided to explore moderation effects by child age, which we believe addresses the reviewers concerns here. 

P18 is it 12 or 13 sessions? The programme length is 13 sessions as stated. 

I think it is a limitation that this only relates to mothers and wonder whether fathers were included in the groups and what data is available on them We agree with the reviewer that parenting research with fathers is very important, and it is indeed a limitation of many studies (including ours) that fathers are often not included. Unfortunately, quantitative data are unlikely to be sufficient in this study to look at fathers and other co-parents. We have, however, now stated that we will: (1) describe the number of other parents involved in the study if possible, and (2) explore the involvement of fathers in the qualitative process evaluation. We have also now described this limitation in our discussion. 

I think it would be helpful to say a little more about the theory of change model We agree with the reviewer that it is important to describe the theory of change to the readers. We believe the clearest way to do so is to provide the IY-T logic model as an additional file, and have now uploaded this as a piece of supporting information. 

Really not sure of the potential of the results to generalise to other programmes. which can be varied in content, process and access. Thank you for noting this important point about generalizability. We have made an amendment in our discussion to clarify that generalizability will depend on the content of individual programs. 

TB declararation

The Children’s Early Intervention Trust is not a training centre for IY programmes and has not been since 2019 

Thank you for noting that the CEIT no longer offer IY training. We have amended our manuscript to state that they previously offered IY training. We feel it is important to keep this competing interests statement in the manuscript, as CEIT will have offered IY training at the time the Barnado’s service started delivering IY-T.

---

## [Editor Report · Decision Letter 1]

1 Sep 2023

A quasi-experimental effectiveness evaluation of the 'Incredible Years Toddler’ parenting programme on children’s development aged 5: A study protocol

PONE-D-23-16914R1

Dear Dr. Mooney,

We’re pleased to inform you that your manuscript has been judged scientifically suitable for publication and will be formally accepted for publication once it meets all outstanding technical requirements.

Kind regards,

Charlotte Lennox

Academic Editor

PLOS ONE
---

## [Editor Report · Acceptance letter]

13 Sep 2023

PONE-D-23-16914R1 

A quasi-experimental effectiveness evaluation of the 'Incredible Years Toddler’ parenting programme on children’s development aged 5: A study protocol 

Dear Dr. Mooney:

I'm pleased to inform you that your manuscript has been deemed suitable for publication in PLOS ONE. Congratulations! Your manuscript is now with our production department. 

Kind regards, 

on behalf of

Dr. Charlotte Lennox 

Academic Editor

PLOS ONE